


# Aerosol data assimilation with aqueous chemistry in WRF-Chem/WRFDA V4.3.1

Soyoung Ha

National Center for Atmospheric Research, Boulder, Colorado, USA

**Correspondence:** Soyoung Ha (syha@ucar.edu)

**Abstract.** This article introduces a new chemistry option in the Weather Research and Forecasting model data assimilation (WRFDA) system coupled with the WRF-Chem model (Version 4.3.1) to activate aqueous chemistry (AQCHEM) for the assimilation of surface concentrations of particulate matter (PM) along with atmospheric observations. The gas-phase mechanism used is the Regional Atmospheric Chemistry Mechanism (RACM), the inorganic aerosols are treated with the Modal Aerosol Dynamics Model for Europe (MADE), and secondary organic aerosol (SOA) production is parameterized based on the Volatility Basis Set (VBS) approach. The "RACM-MADE-VBS-AQCHEM" scheme used in the weakly coupled data assimilation and forecast system facilitates aerosol-cloud-radiation-precipitation interactions through analysis and forecast cycling, accounting for both direct and indirect aerosol effects in the short-term air quality prediction. The new implementation in the three-dimensional variational data assimilation (3D-Var) system was tested with the assimilation of $PM_{2.5}$ and $PM_{10}$ concentrations on the ground over the East Asian region through month-long cycling for Spring 2019. It is demonstrated that the inclusion of aerosol species in the aqueous (or cloud-borne) phase in both analysis and forecast reproduces aerosol wet removal processes in association with the development of clouds, systematically changing the atmospheric composition. The new option with aqueous chemistry in WRFDA is beneficial in air quality forecasting in cloudy conditions, while the simulations without aqueous chemistry overestimate surface $PM_{10}$ ($PM_{2.5}$) by a factor of 10 (3).

## 1 Introduction

Air quality with high particulate matter (PM) concentrations, one of the high-impact weather scenarios affecting our daily lives, is closely tied to weather conditions. Given the short lifetime of aerosol species ($\sim$ one week) and large uncertainties in the atmospheric composition, the optimal initialization of aerosol simulations is key to improving air quality forecasting. Data assimilation (DA) aims to incorporate all available observations into a numerical prediction model to produce initial conditions that lead to the most accurate forecast. When data assimilation is conducted, the quality of the initial condition (or the analysis) hinges on the number of good observations and the model error (e.g., the quality of the forecast model).

Recently, various efforts have been made for aerosol data assimilation, but still using a minimal number of advanced chemistry schemes, especially for the prediction of particulate matter (Chen et al., 2019; Sun et al., 2020; Ha, 2022). Hence, it is fair to say that aerosol data assimilation is still in its infancy, well behind meteorological or oceanographic applications (Baklanov et al., 2014; Bocquet et al., 2015). Major challenges specific to aerosol data assimilation can be described as follows: 1) The



limited information content of the atmospheric composition observations with a lack of accuracy and coverage, especially in cloudy conditions 2) large uncertainties or systematic errors in the chemical transport models, partly due to the significant uncertainties in the forcing parameters such as emissions, and partly due to the imperfection of the parameterized mechanisms that represent all the complex chemical processes and their interactions with the atmospheric environment. 3) Surface PM

concentrations, one of the major contributors to the air quality index, are not prognostically treated in the model. Therefore, the DA system has to deal with dozens of aerosol species in the model that contribute to the estimation of the observed PM variables and determine how to distribute the analysis increments in PM concentrations at the surface back to all the three-dimensional prognostic aerosol variables. As the number of aerosol species in the model is typically much larger than the number of observed variables, the situation always comes down to an under-constrained problem where a unique solution is

not guaranteed. More sophisticated chemical parameterization schemes usually deal with a much larger number of prognostic variables than simple bulk schemes, rendering the aerosol analysis a very high-dimensional problem. The long list of analysis variables to be updated necessitates considerable computational resources, which makes the three-dimensional variational data assimilation (3D-Var) algorithm still attractive, due to its high speed and simplicity for operation, even with its limitations such as static background error covariance and the ignorance of model errors. 4) The model configuration for aerosol chemistry,

the first step toward aerosol cycling, is generally more complicated than that of the weather prediction model. This is because chemical simulations are strongly driven by input forcing data such as anthropogenic, biogenic and/or biomass emissions, as well as weather conditions, to increase the number of input files by several times. In the online coupled system, dynamics and physics configurations for meteorology can greatly affect the reliability and performance of the chemical simulations so that care must be taken in the weather part of the configuration as well. 5) Most of the interfaces to the input data and chemical

processes are highly customized for particular gas and aerosol chemistry schemes because each chemistry parameterization defines a unique set of prognostic variables that are not interchangeable with those in other schemes (Pfister et al., 2020). It means that each chemistry scheme requires a new development of the interface in the variational data assimilation system, including new observation operators (that compute the model correspondents from certain aerosol species) and their tangent linear and adjoint models as well as the background error covariance estimation.

Coupled data assimilation is a growing area of research and development, but it has been mostly confined to global large-scale applications, for example, by coupling the atmosphere with ocean or land (WWRP, 2017). Attempts for developing coupled data assimilation between chemistry and meteorology for regional forecast applications have been limited, although high correlations between the two components have been broadly recognized and realized in the modeling community (Baklanov et al., 2017). In the 3D-Var context, fully (or strongly) coupled data assimilation is still prohibitive because many factors that

come into play in estimating cross-covariances between meteorological and chemical variables are highly variable in time and space, spanning across scales. For real observations, developments and research on coupled data assimilation mostly focused on trace gases such as ozone and carbon monoxide with simplified background error covariances (Ménard et al., 2019). In weakly coupled data assimilation, the coupling occurs during the forecast by using a coupled forecast model, not through the analysis. Observations are assimilated in each component (e.g., meteorology or chemistry) to update the analysis variables

independently, which are then used together to initialize the coupled modeling system for prediction. The background error



covariances are estimated from the forecasts produced by the fully coupled model, reflecting the coupling aspects from the model simulations, but the direct influence of the observations is limited to each component of the model.

Aerosol particles play an important role in a number of key processes related to atmospheric chemistry and physics (Rosenfeld et al., 2008; Stevens and Feingold, 2009; Tao et al., 2012; Baklanov et al., 2014). The atmospheric aerosols directly scatter and absorb incoming solar radiation to directly change the atmospheric radiation reaching the Earth's surface (e.g., "aerosol direct effects"). They are either suspended in the air or attached to hydrometeors such as cloud droplets or ice crystals, serving as cloud condensation nuclei (CCN) or ice nuclei (IN), to alter the formation, lifetime, and optical properties of clouds (such as cloud albedo), and precipitation rates, thereby indirectly impact the radiative transfer (e.g., "aerosol indirect effects"). During cloud processes, aerosol particles undergo physical and chemical changes in their composition and mass concentrations, and are redistributed by clouds and convection that act as transport media over time scales of minutes to hours (Ervens, 2015). The aerosol-cloud interactions, describing both the effects of aerosols on clouds and the cloud effects on aerosol particles, are vital to the daily regional air quality as well as the global atmospheric energy budget. In short-range air quality forecasting, the representation of aerosol-cloud interactions can substantially affect the prediction of particulate matter (PM) concentrations near the surface, especially under the condition of heavily polluted clouds.

From the data assimilation perspective, cloudy conditions are particularly challenging because most retrievals from ground-based or remote sensing instruments are either missing or significantly degraded in their quality due to cloud contamination. When there are no good-quality observations distributed over the domain of interest, the analysis quality heavily relies on the accuracy of the numerical prediction system, especially its systematic errors, as they can severely violate the assumption of unbiased forecast errors commonly adopted in most data assimilation algorithms (Dee and Da Silva (1998); Dee (2005)). Because it is not straightforward to detect and correct a model bias error (and its source), it is imperative to take advantage of all the advanced physical and chemical mechanisms and optimize the model performance (e.g., reducing model errors) as much as possible. Eck et al. (2018, 2020) reported that major air pollution events over East Asia are often associated with significant cloud cover, making aerosol-cloud interactions more important than pristine conditions.

The Weather Research and Forecasting model coupled with Chemistry (WRF-Chem; Grell et al. (2005)) facilitates aerosol-cloud-radiation-precipitation interactions with various combinations of physics and chemistry options for regional applications. To simulate aerosol-cloud interactions or to account for aerosol effects on clouds, aerosol chemistry should be combined with aqueous chemistry so that cloud-borne aerosols (e.g., particles attached to cloud droplets) are explicitly treated as well as interstitial aerosols (e.g., particles suspended in the air). When aqueous chemistry is parameterized with the representation of aerosol particles in the aqueous phase (or in cloud water), cloud droplet number concentrations are prognostically treated through the processes of droplet activation, scavenging, resuspension, and chemical production. And air pollutants transported via convection and wet deposition are also parameterized, leading to the changes in the aerosol size and composition distributions. Wet deposition is the transport and removal of soluble or scavenged constituents by precipitation. It includes in-cloud scavenging and removal by rain and snow (rain-out), release by evaporation of rain and snow, and below cloud scavenging by precipitation falling through without formation of precipitation (wash-out) (Seinfeld and Pandis, 2006). Without aqueous chemistry activated, aerosols cannot directly affect the formation and growth of clouds and cannot be displaced through convec-





tive transport or removed by wet scavenging, which can lead to systematic forecast errors in cloudy conditions. Tuccella et al. (2015) implemented aqueous chemistry (AQCHEM) in the Regional Atmospheric Chemistry Mechanism (RACM; Stockwell et al. (1997)) gas-phase chemistry, coupled with the Modal Aerosol Dynamics Model for Europe (MADE; Ackermann et al. (1998)) inorganic aerosol mechanism and the secondary organic aerosol (SOA) scheme based on a four-bin volatility basis set

(VBS) (Ahmadov et al., 2012) in the WRF-Chem model. They demonstrated that the RACM-MADE-VBS-AQCHEM scheme coupled with cloud microphysics and radiation parameterization schemes could characterize aerosol-cloud feedbacks, reducing large uncertainties in the prediction of microphysical and optical properties of clouds.

The WRFDA system has been recently updated for the RACM-MADE-VBS scheme (chem_opt=108) in the WRF-Chem model for chemical data assimilation (Ha, 2022). Through a case study during the Korea–United States Air Quality (KORUS-

AQ) period, it was demonstrated that the 3D-Var aerosol analysis resulted in systematic improvements in the prediction of surface PM concentrations over Korea but forecast errors tended to increase in cloudy conditions. This study further extends the WRFDA 3D-Var system for the RACM-MADE-VBS-AQCHEM scheme (chem_opt=109) to allow aerosol-cloud interactions with aqueous chemistry in aerosol cycling, with the goal to improve the short-range prediction of surface PM concentrations over South Korea under cloudy conditions. An overview of the WRF-Chem forecast model for aerosol effects is presented in

Section 2, and the new implementation in the WRFDA system, including new forward operators and background error statistics with aqueous chemistry, is presented in Sect. 3, followed by cycling experiments and the forecast verification against independent observations described in Sect. 4. Finally, conclusions are made in Sect. 5, along with a discussion on the limitations of this study and suggestions for future research.

## 2   WRF-Chem for aerosol effects

The WRF-Chem model has long been used to study a wide range of atmospheric phenomena associated with atmospheric chemistry and aerosols over regional domains (e.g., Ntelekos et al. (2009), Grell and Baklanov (2011), Pfister et al. (2011), Ahmadov et al. (2012), Saide et al. (2012), Yang et al. (2015)). The chemical transport model numerically solves the concentration of chemical species through emissions, advection, vertical mixing with dry deposition, convective transport, gas chemistry, cloud chemistry (for activated aerosols in cloud water), aerosol chemistry, and wet scavenging. At the end of every time step,

surface PM concentrations are diagnostically computed for each chemical option.

In WRF-Chem, aerosol direct and indirect effects can be simulated through interactions with atmospheric radiation, photolysis, cloud microphysics, and cumulus parameterization. And aerosol-cloud interactions involve activation, resuspension, aqueous reactions, and wet removal of aerosol particles. To account for aerosol-cloud interactions, the direct effects of aerosols on incoming solar radiation should be first simulated by relating aerosol sizes and compositions to aerosol optical properties

(Fast et al., 2006). The WRF-Chem model provides several different options, but aerosol particles within a size range (e.g., mode) are often assumed to have the same composition so that refractive indices for spherical particles can be simply averaged over all the species for each mode (e.g., volume averaging), as used in this study. Extinction coefficients due to aerosol scattering and absorption are then computed based on Mie theory and passed onto shortwave and photolysis schemes. While aerosol



direct effects can be accounted for by all the aerosol chemistry schemes when either the RRTMG or the Goddard shortwave
radiation is chosen, aerosol indirect effects are only supported by several modal and sectional aerosol options in WRF-Chem.
In cloud microphysics, double-moment schemes such as Lin (Lin et al., 1983) or Morrison (Morrison et al., 2009) should be
employed to predict the number concentrations as well as mass mixing ratios of hydrometeors in the representation of the
particle size distributions. This study uses the Morrison two-moment scheme (Morrison et al., 2009) that predicts the mass
and number concentration of five species (i.e., cloud droplets, cloud ice, snow, rain, and graupel). It is noted that the new
implementation in DA is applicable to any double-moment schemes without any modifications. The microphysics accounts for
the autoconversion of cloud droplets to rainwater based on the droplet number concentrations, and interacts with prognostic
aerosols to alter their size and composition via aqueous processes and wet scavenging (Yang et al., 2011). For subgrid-scale
features, by choosing the Grell-Devenyi cumulus parameterization scheme, atmospheric radiation and photolytic processes can
recognize convective precipitation, and chemical species can be displaced through convective transport. To represent aerosol
effects on clouds and convective transport, aerosol activation should be parameterized so that aerosol particles can grow by wa-
ter condensation to form cloud droplets based on supersaturation and the size of particles (Abdul-Razzak and Ghan, 2002). The
aerosols in the aqueous phase are interacting with cloud microphysics, serving as cloud condensation nuclei (CCN), to affect
cloud droplet number concentrations and cloud radiative properties. And aerosol properties are affected by numerous factors
such as droplet number and mass concentrations, photolytic reactions, chemical composition, gas- and aqueous-phase chem-
istry, dry deposition, wet removal by in-cloud and below-cloud scavenging, water uptake, and resuspension from evaporated
cloud and raindrops.

Tuccella et al. (2015) implemented the RACM-MADE-VBS-AQCHEM scheme (chem_opt=109) for the simulation of
aerosol-cloud-radiation interactions, following Fast et al. (2006) and Chapman et al. (2009), with simple aqueous reactions.
The MADE-VBS aerosol scheme defines the particle size distribution as a superposition of three log-normal modes: an Aitken
mode with a median diameter of 0.01 $\mu$m, an accumulation mode ranging between 0.01 and 1 $\mu$m, and a coarse mode for parti-
cles typically larger than 1 $\mu$m (with a median around 10 $\mu$m). All aerosol particles are assumed to be spherical and internally
mixed (Aquila et al., 2011). The aerosol species treated are sulfate ($SO_4^-$), nitrate ($NO_3^+$), ammonium ($NH_4^+$), elemental car-
bon (EC), primary organic matter (POA), anthropogenic and biogenic secondary organic aerosol (SOA), chloride (Cl), sodium
(Na), unspeciated $PM_{2.5}$, unspeciated coarse fraction of $PM_{10}$ (antha), soil dust (soila), and sea salt (seas). The unspeciated
$PM_{2.5}$ includes the fine fraction of sea salt and mineral dust aerosols.

For aqueous processes, each aerosol species is defined in the aqueous (or cloud-borne) phase as well as in the interstitial (or
non-activated) state. The number and mass concentrations of activated aerosols are calculated for each mode in the presence of
water supersaturation. In this study, several bugs were found and fixed in the RACM-MADE-VBS-AQCHEM scheme in WRF
V4.3.1, primarily related to the production of sulfuric acid.





## 3 WRFDA modifications for aqueous chemistry


An interface between the WRF-Chem model and the WRFDA 3D-Var system in version 4.3.1 is extended for the RACM-MADE-VBS-AQCHEM option to facilitate the assimilation of $PM_{2.5}$, $PM_{10}$, $SO_2$, $NO_2$, $O_3$, and CO measurements on the ground. The RACM-MADE-VBS scheme without aqueous chemistry (chem_opt=108) was implemented in Ha (2022), and this study built an interface for the same chemistry including aqueous-phase aerosols (e.g., chem_opt=109) to reflect aerosol-cloud

interactions in the aerosol analysis.

### 3.1 Cost function

In the three-dimensional variational data assimilation (3D-Var) system, the cost function $J(\mathbf{x})$ is minimized to find an optimal solution for the model state ($\mathbf{x}$) that best fits to all the observations ($\mathbf{y}$) available at the analysis time based on background and observation error covariance matrices ($\mathbf{B}$ and $\mathbf{R}$, respectively) under the assumption of the Gaussian error distributions

(Lorenc, 1986). In the incremental formulation (Courtier et al., 1994) adopted in WRFDA, analysis increments $\delta\mathbf{x}(= \mathbf{x} - \mathbf{x_b})$ are computed at each iteration using the background forecast ($\mathbf{x_b}$) from the previous analysis ($\mathbf{x_a}$) or the previous iteration step, and the control vector ($\mathbf{v}$) is defined as $\delta\mathbf{x} = \mathbf{B^{1/2}v}$ to construct the cost function as below.

$$J(\mathbf{v}) = \frac{1}{2}\mathbf{v^T v} + \frac{1}{2}(\mathbf{d} - \mathbf{HB^{1/2}v})^T\mathbf{R^{-1}}(\mathbf{d} - \mathbf{HB^{1/2}v}), \tag{1}$$

where the innovation vector is defined as $\mathbf{d} = \mathbf{y} - \mathbf{H}(\mathbf{x_b})$ and the observation operator $\mathbf{H}$ transforms the model states ($\mathbf{x}$)

to the observed quantities ($\mathbf{y}$) at observation locations. In chemical or aerosol data assimilation presented in this study, all the chemical species defined in the model states ($\mathbf{x}$) are also used as control variables ($\mathbf{v}$) with univariate error covariances (e.g., assuming no static cross-correlations between chemical species or between chemical and meteorological variables). A list of 32 three-dimensional aerosol species defined in the analysis is aerosol sulfate (so4ai and so4aj), nitrate (no3ai and no3aj), ammonium (nh4ai and nh4aj), chloride (clai and claj), primary organic matter (orgpai and orgpaj), elemental carbon

(eci and ecj), sodium (naai and naaj), unspeciated $PM_{2.5}$ (p25ai and p25aj), 4-bin anthropogenic and biogenic SOA (asoa1i, asoa1j, asoa2i, asoa2j, ..., bsoa4i, bsoa4j), where each variable name in the parenthesis ends with $i$ or $j$ to indicate Aitken or accumulation mode. Also included are three coarse-mode variables - non-reactive anthropogenic aerosol (antha), marine aerosol concentration (seas), soil-derived aerosol particles such as dust (soila). Total of 35 aerosol species in all three modes are also defined in the aqueous (or cloud-water) phase, with 'cw' added in the name. In case four gas species ($SO_2$, $NO_2$, $O_3$,

and CO) are assimilated, they are also included so that the total number of three-dimensional chemical species becomes 74 in the new WRFDA system.

### 3.2 Forward operator and observation errors

In the assimilation of surface PM observations, $\mathbf{H}$ is a sum of each aerosol species defined in the control vector ($\mathbf{v}$), interpolated at each observation site, following the way the MADE-VBS aerosol scheme in the WRF-Chem model estimates PM concen-

trations based on individual aerosol species. For the activation of aqueous chemistry, $\mathbf{H}(\mathbf{x})$ is extended to include cloud-borne





(activated) as well as interstitial (non-activated) aerosol species in all three modes. The $PM_{2.5}$ concentrations in the model space ($\mathbf{y}_{pm_{2.5}}$) are computed as the sum of all the aerosol species listed above in accumulation (j) and Aitken (i) modes.

$$\mathbf{y}_{pm_{2.5}} = \rho_d \sum_{p=1}^{N} (\sum_{m=i}^{j} \mathbf{y}_m^p + \sum_{m=i}^{j} \mathbf{y}_m^{*p}), \tag{2}$$

where N is 32 and $\rho_d$ dry density ($[kg/m^3]$) for the unit conversion from aerosol mixing ratios ($\mu g/kg$) to mass concentrations
($\mu g/m^3$), $\mathbf{y}_m^p$ and $\mathbf{y}_m^{*p}$ each species in interstitial and aqueous phase, respectively.

When $PM_{10}$ is assimilated alone, the model correspondent is computed by adding three coarse-mode variables - anthropogenic primary aerosol (antha), marine aerosol concentration (seas), soil-derived aerosol particles such as dust (soila) - into the simulated $PM_{2.5}$. For the aqueous chemistry option, the three coarse-mode variables in the aqueous phase (anthcw, seascw, and soilcw) are also included in the observation operator. If $PM_{10}$ is assimilated with $PM_{2.5}$, the residuals from ($PM_{10}$ - $PM_{2.5}$)
are assimilated as a sum of three coarse-mode aerosols, following Ha (2022).

The assimilation of trace gases is straightforward because each gas species is specified in the model prognostic variable. The control variables are the same four gas species ($SO_2$, $NO_2$, $O_3$, and CO), and the observation operator becomes a simple horizontal interpolation of the corresponding variable at the lowest model level.

It is also noted that three-dimensional aerosol number concentrations are not directly associated with PM mass concentra-
tions and there are no such routine observations, aerosol number concentrations in each mode and in each phase are thus not included in the analysis variables and the observation operators. Therefore, it may not be appropriate to examine the effects of aerosol-cloud interactions on the changes of aerosol number and size distributions within the current WRFDA system.

The observation error covariance matrix $\mathbf{R}$ remains the same regardless of aqueous chemistry, using the same uncorrelated observation errors for each observation ($y_o$). Following Ha (2022), the observation error for surface PM consists of the mea-
surement error ($\epsilon_o$) and the representative error ($\epsilon_r$): $\epsilon_y = \sqrt{\epsilon_o^2 + \epsilon_r^2}$ where $\epsilon_o = 1.5 + 0.0075 * y_o$ and $\epsilon_r = \gamma \epsilon_o \sqrt{\frac{\Delta x}{L}}$. Here, $\gamma$ is set to be 0.5, $\Delta x$ is grid spacing (here, 27 km for domain 1 and 9 km for domain 2) and the scaling factor L is defined as 3 km.

For the system reliability, data quality control (QC) is done by setting maximum thresholds of observation values and innovations ((o - f)'s) during the assimilation procedure. Surface $PM_{2.5}$ and $PM_{10}$ observations are rejected when they are
greater than 300, 500 $\mu g$ m$^{-3}$, respectively, or are different from their model equivalent (e.g., $\mathbf{H}(\mathbf{x_b})$) by more than 100 $\mu g$ m$^{-3}$. In terms of gas species, the maximum threshold is set to be 2, 2, 2, and 20 ppmv for the observed $SO_2$, $NO_2$, $O_3$, and CO, respectively. They are also rejected based on the thresholds of 0.2, 0.2, 0.2, 20 ppmv for their innovations, respectively. In the current implementation, gas-phase pollutants on the ground are designed to be assimilated together, as opposed to individual species, using the corresponding model variables as their analysis (or control) variables.



## 3.3 Background error covariance

To avoid the inverse calculation of the large $\mathbf{B}$ matrix, the square root of the $\mathbf{B}$ matrix $(\mathbf{B} = \mathbf{B^{1/2}}(\mathbf{B^{1/2}})^{\mathbf{T}})$ is decomposed into a series of sub-matrices in the WRFDA system.

$$\mathbf{B^{1/2}} = \mathbf{U_p S U_v U_h} \tag{3}$$

where the $\mathbf{U_p}$ matrix is called physical or balance transformation (via linear regression), $\mathbf{S}$ a diagonal matrix of forecast error standard deviation, $\mathbf{U_v}$ the vertical transform, and $\mathbf{U_h}$ the horizontal transform matrix.

In this study, the WRF-Chem model is configured with two domains at 27 and 9 km grid resolutions, respectively, in a one-way nesting mode, as illustrated in Fig. 1a. Vertically, total of 31 model levels are configured up to 50 hPa, with upper level jets are located around the model level 20 while the low level jets (LLJ) are situated around level 7. Chemical simulations are carried out starting at 00 UTC every day from 21 Feb to 21 Mar 2019 to compute background error covariance statistics for chemical species defined in the RACM-MADE-VBS-AQCHEM scheme. Differences between 24 and 48 h forecasts at the same validation time are then used as a proxy for forecast errors in each domain, and a total of 28 sample forecasts were used to construct the B matrix through the National Meteorological Center (NMC) method (Parrish and Derber, 1992). The GENBE 2.0 software (Descombes et al., 2015) is also expanded to accommodate all the aerosols in cloud water in the B matrix. As in the previous study (Ha, 2022), all the grid points are binned together for each model level, with no latitudinal or longitudinal dependencies in the background error covariance. To examine the changes due to aqueous chemistry in the B matrix, the same one-month cycling forecasts are conducted with and without aqueous chemistry, called NO_AQ and AQ, respectively.

Figure 2 compares the background error covariance ($\mathbf{B^{1/2}}$) between the two experiments in each aerosol component in the interstitial state. As the control vector is multiplied by $\mathbf{B^{1/2}}$ to convert to the increments in the control variables ($\delta\mathbf{x}$) after the assimilation, this figure shows the relative weights for each species that decide how much they can contribute to atmospheric constituents and where they can contribute most in vertical when the observed PM concentrations on the ground are assimilated. With aerosols in cloud water accounted for in the simulations (AQ), coarse mode aerosols experience the largest changes (in the bottom panels), followed by accumulation mode aerosols (ending with "aj"). Sulfate aerosols in the interstitial state become less important in both accumulation (so4aj) and Aitken (so4ai) modes, with larger reductions going down to the surface. On the contrary, nitrate and sodium aerosols in accumulation mode (no3aj and naaj, respectively) contribute more in the low troposphere with aqueous chemistry. Increases in ammonium, unspeciated P25, primary organic aerosols and elemental carbon aerosols - nh4aj, p25j, orgpaj, ecj, respectively - are confined to the boundary layer (e.g., within the lowest five model levels), and their companions in the Aitken mode change only by one order magnitude smaller. The most significant changes are observed in the accumulation mode chloride aerosol (claj) at the surface, reduced to 1/6 of the value in NO_AQ. It is not clear why the atmospheric constituents at the surface were overwhelmingly dominated by claj with no aqueous chemistry, but they seem to be better balanced out between the species when aqueous chemistry is activated. While coarse mode sea salt (seas) and dust aerosols (soila) are reduced in the entire troposphere, coarse mode anthropogenic aerosols (antha) are redistributed downward with significant increases near the surface when aqueous chemistry is used. Secondary organic aerosols are mainly changed in the accumulation mode (SOAj), generally reducing in the upper troposphere and increasing near the surface.





When aqueous chemistry is turned on (AQ), aerosols in cloud water ("cw") are also included in the B matrix, mostly confined
below the model level 15 (e.g., mid-to-low troposphere), as shown in Fig. 3. Aqueous-phase aerosols are largely distributed in
the lower troposphere with the maximum around level 5, in association with low-level clouds, while most interstitial aerosols
(depicted in Fig. 2) tend to increase as the level goes down, producing the maximum at the surface. The most distinctive
changes with AQ are found in sulfate and sea salt aerosols. In the accumulation and Aitken modes, the largest error standard
deviation is presented in sulfate aerosols (so4cwj and so4cwi in brown solid lines in Figs. 3 (a) and (b), respectively), followed
by nitrate and ammonium aerosols (orange and red dotted lines with 'x' symbol, respectively). Coarse mode aerosols in cloud
water are dominated by sea salt (SEAS; blue dotted line in Fig. 3 (c)) around the same level 5 while dust aerosols (SOILA;
cyan) show the peak around level 10, which might be related to long-range transport in the higher levels.

## 4 Chemical analysis and forecast cycling

The WRF system consists of the WRF preprocessing system (WPS), the weather prediction model coupled with Chemistry
(WRF-Chem), and its data assimilation (WRFDA). Rendering analysis variables consistent with forecast variables in the model,
one can construct a unified system for air quality forecasting, where the air quality prediction is not only taking advantage of
real-time interactions between meteorology and chemistry but also is initiated from its own analysis based on its forecast error.
Through cycling (e.g., conducting analysis and forecast consecutively) at certain time intervals (ex. every 6 h), the observed
information is continuously incorporated into the WRF-Chem model to better initialize the simulations. By pulling out the
model trajectory towards observations every cycle, we systematically help the model produce atmospheric constituents close
to the observed information.

Figure 4 shows a flowchart of the WRF-Chem/WRFDA cycling system with chemical data assimilation. Dotted lines imply
optional input data while solid lines the mandatory inputs for WRF-Chem/WRFDA cycling, accompanied by typical input file
names (with no specification of domain ID or time) used in the WRF system. As the fist step, WPS is run to configure the
model domain with geological data such as landuse and topography (geogrid.exe), ungrib meteorological data (ex. the UK Met
Office analysis; UM MET) in the grib format (ungrib.exe), and transform the three-dimensional data into the WRF domain
(metgrid.exe). Through the WRF initialization step (real.exe), the data is then converted to the initial condition (wrfinput) and
lateral boundary condition (wrfbdy) files for meteorological variables in each domain. For chemical simulations, emissions data
should be prepared based on the wrfinput file to define land use categories consistent with those used for the meteorological
initial condition. As soon as the WRF-Chem model starts, atmospheric physics and chemistry parameterizations are initialized
based on the land use categories (e.g., mminlu) in the look-up tables such as LANDUSE.TBL and VEGPARM.TBL. It is thus
critical to use the same wrfinput file in producing all the emissions data. By default, WRF-Chem regional simulations use
an idealized gas profile for some chemical species at lateral boundaries. In order to replace them with realistic time-varying
chemical boundary conditions (chembc), a utility named "mozbc" is provided, along with other pre-processing tools, to process
global chemistry model output files (https://www2.acom.ucar.edu/wrf-chem/wrf-chem-tools-community/, last access: 22 April
2022) for the region of interest. It should be noted that the diagram shown here is not meant to describe all the possible data



input in the WRF-Chem model. Optional input data such as an upper boundary condition for some gas species, biomass burning (e.g., fire) or aircraft emissions data are not included because they were not considered in the present study.

Without data assimilation, wrfinput ($x_b$) and wrfbdy files are directly used to initialize the model simulation, skipping
WRFDA processes (e.g., da_wrfvar.exe and da_update_bc.exe). But if one wants to update the initial condition in the variational data assimilation, at least three input files are required for each model domain - a first guess ($x_b$; wrfinput or fg), background error covariance ($B$; be.dat), and observations ($y$; ob.ascii or ob.bufr) that usually come with the specification of observation errors ($R$). Before incorporating observations into the DA system, data collection and processing should be carefully carried out, including data quality check. Since WRF Version 4 (including WRFDA), simultaneous data assimilation
has been available for a few chemical options in WRF-Chem to update meteorological and chemical fields at the same time. In the current implementation, chemical observations are designed to be available in ascii format (ob_chemsfc.ascii), separate from meteorological (MET) data provided in BUFR format. When data assimilation (da_wrfvar.exe) is run for each domain, the initial condition is updated as the analysis ($x_a$) in each domain. The lateral boundary condition in the mother domain also needs to be updated (through da_update_bc.exe) to be consistent with the analysis in the boundary zone.

To compute the background error covariance ($B$), WRF-Chem forecasts should be run in advance, typically cycling without data assimilation using the same model configuration for a long period of time (at least for one month). In the NMC method, forecast differences between two different forecast leads at the same validation time are used to estimate the background error covariance for all the analysis variables in each domain using the GENBE software.

Once the WRF-Chem model is integrated from the initial condition, the output forecast reached the next cycle is reused to
provide the next first guess with the simulated chemical species (e.g., wrf_chem_input). By repeating the WRF initialization, WRFDA, and WRF-Chem simulations with the recycled chemical species at the cycling frequency, WRF-Chem/WRFDA cycling can be carried out continuously.

## 4.1 Cycling experiments

The WRF-Chem/WRFDA cycling experiments are conducted over the East Asian region (with 27-km grid resolution) nested
down to the Korean peninsula (at 9-km grids) for a month-long period (Feb 21 - Mar 23, 2019) every 6 h. This study uses the Morrison two-moment scheme (Morrison et al., 2009) for cloud microphysics, Grell-3 for cumulus parameterization (Grell and Dévényi, 2002), the YSU scheme (Hong et al., 2006) for the planetary boundary layer (PBL), and the rapid radiative transfer model for GCMs (RRTMG) for both shortwave and longwave radiation (Iacono et al., 2008). The direct aerosol effect is accounted for through interactions with atmospheric radiation and photolysis while indirect aerosol effects are represented
through interactions with cloud microphysics. Dust and sea salt emissions are simulated online, following the GOCART mechanism (e.g, dust_opt = 13 and seas_opt = 2). Photolysis rates of chemical species are computed in a simplified version of the National Center for Atmospheric Research (NCAR) Tropospheric Ultraviolet-Visible (TUV) model (named the Madronich scheme; phot_opt=1) (Madronich, 1987). A list of physics and chemistry schemes used in this study is summarized in Table 1.

Anthropogenic emissions data for the RACM-MADE-VBS scheme are obtained from the National Institute of Environment
Research (NIER) that is in charge of operating daily air quality forecasting in South Korea. The data consists of a single level



with no plume rise or any vertical distribution. Biogenic emissions are built up online using the Model of Emission of Gases and Aerosol from Nature (MEGAN; Version 2) (Guenther et al., 2006), but biomass burning emissions are not used in this study. All the WRF files including anthropogenic and biogenic emissions are processed based on the MODIS land use datasets (Friedl et al., 2002).

The initial and lateral boundary conditions for meteorological variables are produced by global forecasts from the UK Met Office's Unified Model (UM) operated by the Korean Meteorological Administration (KMA) every 6 h. Chemical or upper boundary conditions are not used in this study.

Hourly surface observations of $PM_{2.5}$ and $PM_{10}$ are collected from 279 South Korean sites operated by AIRKOREA (http://www.airkorea.or.kr, last access: 27 April 2022) and 765 Chinese sites from the China National Environmental Mon-
itoring Center (CNMEC; http://www.cnemc.cn, last access: 27 April 2022) over the model domain. As the measurements are highly concentrated in the large cities, the Korean sites are randomly split for assimilation and verification, respectively, and each dataset is then averaged over the 9-km model grid. As a result, 279 Korean sites are processed into 219 stations for assimilation while the other 100 sites are averaged to 71 independent observations for evaluation over South Korea. The same Chinese data are used for both assimilation and verification without such data processing because the main focus of this study
is to examine the aerosol impact over South Korea. Figure 1 depicts the surface network used for assimilation (in (a)) and for evaluation (in (b)), respectively. For meteorological data assimilation, all the conventional observations in the National Centers for Environmental Prediction (NCEP) prepbufr data (https://rda.ucar.edu/datasets/ds337.0/; last access: 27 April 2022) are employed. For the verification against weather observations, a total of 699 surface Automatic Weather System (AWS; https://www.weather.go.kr/weather/observation/aws_table_popup.jsp, last access: 27 April 2022) sites in South Korea (blue
dots in Fig. 1c) are employed.

As summarized in Table 2, two baseline experiments are performed with and without aqueous chemistry (NODA and NODA_AQ, respectively). Based on the background error statistics computed from each of the experiments, two corresponding DA cycling runs are then conducted with the same model configuration, assimilating surface $PM_{2.5}$ and $PM_{10}$ concentrations (DA and DA_AQ, respectively).

Figure 5 compares the analysis and background profiles of each aerosol component, averaged over 71 verification sites over South Korea, in two DA experiments (DA and DA_AQ) from 6 hourly cycles for 1-23 March 2019. In the 3D-Var analysis, analysis increments in PM concentrations are distributed across aerosol species based on the background error covariance, the vertical structure of each species generally follows their background error structures illustrated in Figs. 2 and 3. Since Aitken mode aerosols are very small compared to their companions in the accumulation mode, they are combined for each aerosol
component. Although only surface concentrations are assimilated, the impact goes up to the upper troposphere, again based on the forecast errors of each component. Accumulation and Aitken mode aerosols ((a)-(h)) show most changes confined to the boundary layer, with an increasing trend of concentrations by both AQ and DA except for NaCl in (h) where the extreme surface chloride value disappears with AQ. Unlike smaller particles, coarse mode aerosols ((i)-(k)) are systematically reduced by AQ all the way up to level 20 (around 200 hPa), implying that aerosol sizes get smaller when accounted for in clouds.





Figure 6 compares two DA experiments in a box plot of innovations ((o-b)'s) categorized for different events based on Table 3. As it goes with higher pollution events, innovations in both experiments gradually increase in terms of mean, median and standard deviation over Korean sites, indicating larger errors with larger uncertainties. Both simulations tend to overestimate good or moderate events (o < b) while underestimating unhealthy or very unhealthy conditions (o > b) in both $PM_{2.5}$ and $PM_{10}$ concentrations. Data assimilation with aqueous chemistry (DA_AQ) slightly improves the fit to observations in all events

except for the modest underestimation in surface $PM_{10}$ in unhealthy (80 - 150 $\mu g\ m^{-3}$) and very unhealthy (> 150 $\mu g\ m^{-3}$) conditions. Whiskers (corresponding to the tails of the probability density function) get longer with higher pollution events, implying that the spatial variability of surface PM concentrations goes up with haze events that might not be fully captured by the 9-km simulations.

## 4.2   Air pollution events in cloudy conditions

Air pollutants transported to the Korean peninsula are susceptible to the moist environments above the Yellow sea between China and Korea, and the extent to which aerosol particles interact with moisture or cloud droplets is subject to the moving speed and direction of the synoptic weather systems (such as fronts or troughs) that pass over the sea. To examine the effect of data assimilation with aqueous chemistry, an air pollution case with substantial cloud cover is chosen during the cycling period. Figure 7 illustrates Level 3 (gridded) daily mean $1°$ aerosol optical depth (AOD) retrieved from the Visible

Infrared Imaging Radiometer Suite (VIIRS) onboard the Suomi National Polar-Orbiting Partnership (Suomi NPP) spacecraft (https://ladsweb.modaps.eosdis.nasa.gov/missions-and-measurements/viirs/, last access: 27 April 2022) on 19 March 2019. The total column AOD indicates high aerosol loading over Korea, especially along the west coast near the Seoul Metropolitan Area. A surface high pressure center is located southwest to Jeju island (around $32°N$, $125°E$) and the anticyclonic circulation over the sea forms south-westerlies to bring air pollutants and moisture into South Korea (not shown). The areas with no colors

in the satellite image are commonly involved with cloud contamination that leads to missing data.

    On the next day, with a surface low pressure system moving from eastern China, significant cloud cover is observed over a wide range of the region. And the low cloud top pressure retrieved from MODIS sensors onboard the Aqua satellite (DOI: 10.5067/MODIS/MYD06_L2.061) in Fig. 8 indicates that convective clouds (in white) are developed over the Yellow sea near Seoul, South Korea. Such a cloudy condition makes most remote-sensing retrievals missing or of poor quality over the entire

region, leaving in-situ surface measurements as major resources of the observed information.

    In association with the extensive cloud cover and long-range transport embedded in the synoptic atmospheric flows, air quality exhibits large variations for the next few days. Figure 9 compares surface $PM_{2.5}$ (top) and $PM_{10}$ (bottom) concentrations in 24 h forecasts from the analysis at 00 UTC every day for three consecutive days (19-21 March 2022). Each point (for both observations and forecasts) represents the averages over 71 verification sites (red dots in Fig. 1b). On the first day of 19 March

2019, the 00Z analyses show that both DA (blue square) and DA_AQ (red circle) produce slight overestimation (by up to 9 and 18 $\mu g\ m^{-3}$ in $PM_{2.5}$ and $PM_{10}$, respectively) while NODA experiments (with or without AQ) underestimate $PM_{2.5}$ and $PM_{10}$ by up to -12 and -15 $\mu g\ m^{-3}$, respectively, implying that data assimilation generally escalates surface PM concentrations to compensate for the model's underprediction. As the day goes on, air quality is observed to get worse gradually and modestly,





but all experiments predict a steep increase of both PM concentrations with time, leading to considerable overestimation at
night (from 12 to 23 UTC). It is noted that only DA_AQ forecasts reduce PM concentrations after 12 UTC (e.g., 21 KST in
local time) to progressively match the observations until the next morning.

On the second day, a time series of mean sea level pressure ([hPa]; gray dotted line in the top panel) is indicative of the pas-
sage of the surface low pressure system accompanied by high cloud cover. And light precipitation ([mm]) is reported overnight
(for 10-15 UTC; 19-24 KST), as indicated by a gray dotted line in the bottom panel. The most distinctive differences are
found after the rainfall event (e.g., 12-23 UTC) depending on whether aqueous chemistry (AQ) is employed. With AQ, fore-
casts can remarkably capture steady decreases in PM observations due to wet scavenging, regardless of DA (e.g., DA_AQ and
NODA_AQ; red circle and orange cross symbols, respectively). On the contrary, DA and NODA predict substantial increases
in both PM concentrations, opposite to the observed trends, likely because the model without AQ has no way to represent
aerosol particles in clouds and the convective transport so that they are all simulated as suspended in the air without any loss
through wash-out or rain-out.

Such a large model error affects the analysis quality at the next cycle, thereby the errors of the following 24 h forecasts. On
21 March 2019, the errors at 00 UTC are significantly different between experiments, especially in $PM_{10}$ concentrations. DA
(blue square) shows large improvements over NODA (green "x"), but still produces almost triple the observed value (72 vs.
27 $\mu g\ m^{-3}$). With data assimilation, PM forecasts tend to be slightly overestimated, but the use of AQ still improves forecasts
up to 18 h. After 18 UTC, all experiments miss the slight increase in PM concentrations. It is noted that the poor quality of
PM forecasts over nighttime is in part because of the coarse vertical resolution employed in this study, which was set to follow
the operational configuration at the NIER. A total of 31 levels with the lowest model level located around 173 m might be too
coarse to simulate the shallow boundary layer at night as the stable boundary layer can go down below 100 meters in this area
of sea level pressure. It would be nice to examine the effect of higher vertical resolutions near the surface in future studies.

To examine how the large differences in surface PM concentrations are associated with cloud and rain droplets, hourly
forecasts of $PM_{10}$ concentrations at Seoul (37.5N, 127E) are compared between experiments on the second day (20 March
2019) in Fig. 10. In WRF-Chem model V4.3.1, as aerosols can only act as cloud condensation nuclei, not as ice nuclei,
aerosol-cloud-precipitation interactions can be directly simulated only for warm rain processes. Also, the background error
covariance defines most of the aerosol impacts below the model level 20 (as shown in Figs. 2 and 3), it is desirable to focus
on the vertical structure of day 2 forecasts up to level 20. Cloud water and rain water mixing ratios are superimposed as white
and pink contours, respectively, and liquid water path (LWP = $\int_0^z \rho q_c dz$, where $\rho$ stands for dry density, $q_c$ cloud water mixing
ratio, $dz$ height differences between two adjacent levels) is overlaid as a black dashed line with y-axis on the right. In the
comparison, it is evident that the use of aqueous chemistry can play an important role in the formation and development of
clouds and wet deposition. Without AQ and DA (Fig. 10 (a)), the simulation is initiated with high concentrations of $PM_{10}$ in
the boundary layer at 00 UTC (09 KST), which remains for most of the day (until 08 UTC; 17 KST). Clouds start to form
around level 15 in the late afternoon and undergo some autoconversion and accretion processes (pink contours) but mainly
persist through the night, moving down to the ground. Autoconversion is a process where cloud droplets collect each other and
become raindrops, while accretion denotes the collection of cloud droplets by falling raindrops. With the development of low



clouds (that do not see or interact with aerosols), air quality is predicted as very unhealthy conditions (surface $PM_{10} > 150$ $\mu g$
$m^{-3}$) for the next few hours from 18 UTC (03 KST). Data assimilation can effectively mitigate the overestimation of low-level
PM concentrations from the initial time to the late afternoon but cannot compensate for the model error due to the lack of
the physics mechanism in the later forecast lead times where a similar structure of high PM concentrations is simulated (Fig.
10 (b)). The activation of aerosols in clouds, however, drastically changes the model behavior to enhance rain water mixing
ratios in the mid-troposphere from the late afternoon (around 08 UTC) and activate wet scavenging of aerosol particles for the
next 6 h or so (Fig. 10 (c)). In DA_AQ (Fig. 10 (d)), data assimilation suppresses the PM overestimation again, but other than
that, the vertical structure and temporal evolution is similar to the one without DA, ensuring that air quality forecasting can
be largely affected by aqueous chemistry interacting with cloud microphysics and wet deposition processes in the model. As
cloud droplet is an important medium for aqueous-phase reactions, an accurate simulation of LWP is crucial. The time series
of LWP depicts that the simulated LWP can be as large as three times when aerosols in clouds are not represented. And the
enhanced mid-to-low level clouds with little autoconversion of cloud to rain water droplets are closely tied to the overpredicted
PM concentrations. It is apparent that clouds can greatly alter the evolution of aerosols in the atmosphere and aerosols can also
exert large influences on the cloud formation and amount.

On the other hand, a time series of PBL height indicates that experiments produce very different evolution of the boundary
layer at night, in association with the development of clouds and rain (not shown). During the daytime, PBL height gradually
increases from the initial time (09 KST) to around 1.2 km in all experiments, but after 08 UTC (17 KST), PBL height mono-
tonically decreases in the aqueous experiments while it grows further to the peak (∼1.5 km) around 11 UTC (20 KST) without
aqueous chemistry. It is evident that aerosols in the aqueous phase can significantly affect the boundary layer vertical mixing
and surface conditions. At 12 UTC, for example, 2-m temperature is underestimated by up to 1 degrees in AQ experiments,
half degrees colder than those without AQ, whereas 2-m relative humidity is closer to observations (92 %) by 5 % with AQ.
It is apparent that surface winds are not very sensitive to the use of aqueous chemistry in this particular study, partly related
to large model uncertainties in the nocturnal stable boundary layer. As Saide et al. (2015) pointed out, aerosols can play an
important role in modifying severe weather conditions or outbreaks. But in the weakly coupled DA system used in this study,
aerosol and weather data assimilation only indirectly affect each other through aerosol feedbacks in the forecast model, and the
assimilation of surface weather observations is not effectively performed owing to the specification of large observation errors.
A thorough investigation of the influence of aerosol data assimilation on meteorological conditions and the optimization of
weather data assimilation is left behind for future studies.

On 00 UTC 21 March 2019, the surface low pressure center accompanying the wide cloud bands is moved to the East Sea
and the air is cleared out in conjunction with wet deposition of aerosols over South Korea. Figure 11 shows that surface $PM_{2.5}$
and $PM_{10}$ measured at most Korean sites are lower than 20 and 50 $\mu g$ $m^{-3}$, respectively.

With no aqueous chemistry, however, surface PM concentrations are significantly overpredicted across domain 1, especially
over the sea surrounding the Korean peninsula, as depicted in Figs. 12 and 13. When AQ is employed in the analysis and
forecast (in the bottom panel), both $PM_{2.5}$ and $PM_{10}$ concentrations over Korea are accurately predicted as very low. In DA,
surface $PM_{2.5}$ ($PM_{10}$) in the Seoul Metropolitan Area, for example, is predicted above 75 (180) $\mu g$ $m^{-3}$ while the prediction





at the same location in DA_AQ is below 15 (30) $\mu g\ m^{-3}$. The difference between the two experiments corresponds to the
prediction of air quality changing from a very unhealthy to a very good condition, which is almost comparable to a situation
of rain or no rain. This suggests that wet deposition and convective transport with aqueous chemistry should be included to
correctly simulate the evolution of aerosols, especially when associated with clouds and precipitation.

It is noteworthy that, in the presented case study, the total accumulation is not very sensitive to the use of aqueous chemistry
or data assimilation. In this particular case with the high cloud cover producing light precipitation, aerosol-cloud-precipitation
interactions affect both chemical and meteorological environments, substantially altering aerosol concentrations, the formation
and development of clouds, and even the boundary layer structure (not shown), but the total rain accumulation varies across
experiments only by a few mm for the day. While the precipitation certainly removes most of the aerosols in the atmosphere
through wet scavenging, the role of cloud-borne aerosols on precipitation is not very clear in this particular case with light
precipitation. A thorough investigation of all the physics mechanisms behind the coupled modeling system is beyond the scope
of this study and left for future studies.

Once cycling experiments are carried out, it is essential to understand the effect of data assimilation for the whole period. The
time mean analysis and background (e.g., 6-h forecast), after discarding the spin-up period of Feb 2019, in Figs. 14 (a) and (b)
summarize the systematic changes in the vertical structure of $PM_{2.5}$ and $PM_{10}$ due to data assimilation and aqueous chemistry.
The DA analysis without aqueous chemistry (orange) shows large surface PM concentrations with an abrupt increase at the
lowest model while the DA_AQ analysis (red dotted line) produces a smooth increase towards the surface with larger $PM_{2.5}$
and $PM_{10}$ concentrations in the boundary layer. However, on 00 UTC 21 March 2019, after the rainfall, DA_AQ markedly
reduces the PM concentrations over the entire low-mid troposphere in both background and analysis (bottom panel), due to
wet deposition. The analysis in DA without AQ (orange) tends to reduce the overestimation in the background, especially in
$PM_{10}$, but still much larger than the background in DA_AQ, meaning that data assimilation cannot fully compensate for the
model error. Main points of these results are threefold: 1) It is hard to make data assimilation effective when the model error is
large 2) Reducing model uncertainties (by advancing physics mechanisms) is one of the most efficient ways of improving the
analysis. 3) It is of great benefit to the forecast quality to cycle analysis and forecast for a long period of time.

The equilibrium between the gas and aqueous phase varies across aerosol species depending on their solubility. The ac-
tivation of aerosols is determined based on the hygroscopicity of each aerosol component, and wet deposition is applied to
individual aerosol species. Hence, the processes of aerosol activation, scavenging, and chemical reactions through aqueous
chemistry can lead to changes in the aerosol size distribution and the atmospheric composition.

And data assimilation of particulate matter concentrations can also shift aerosol particle composition based on the back-
ground error covariance estimated for individual aerosol species. The composition also varies across particle size ranges (e.g.,
modes), but for simplicity, we can focus on the change of atmospheric composition due to data assimilation and aqueous chem-
istry, regardless of the mode. Fig. 15 summarizes aerosol concentrations of each aerosol component in the analysis (gray bar)
and the analysis increment (pink bar) at the lowest model level, averaged over 71 verification stations over South Korea. The
top panel shows the mean of 6-h cycles for 1-23 March 2019 while the bottom panel illustrates the 00Z analysis on 21 March
2019. The pie chart embedded in each panel represents the atmospheric composition in the analysis. As no measurements are




available to evaluate the composition with all the simulated species, these charts are not provided for the purpose of forecast
accuracy, but simply to examine the impact of data assimilation and aqueous chemistry. Comparing with and without aqueous
chemistry (DA_AQ and DA, respectively), it is evident that accounting for aerosols in the aqueous phase largely changes the
aerosol composition in the mean analyses and the analysis increments (e.g., Analysis - Background; AmB). Without aqueous
chemistry (Fig. 15 (a)), the analysis and the increments are dominated by chloride aerosols, consistent with Fig. 5, but the
inclusion of aqueous chemistry tends to balance out across species (Fig. 15 (c)). At 00 UTC 21 March 2019 (in the bottom
panel), after wet scavenging occurs overnight, aerosol concentrations and atmospheric compositions show stark differences
depending on whether or not aqueous chemistry is used. By this time, air pollutants were already washed out, and the analysis
increments (pink bars) make little changes in aerosol concentrations (Fig. 15 (d)). Without aqueous chemistry, the analysis is
predominantly characterized by coarse-mode sea salt aerosols, even with the large decrease in the analysis increments. This
feature might be related to the experimental sea salt option used in this study and needs to be revisited in the future.

## 5   Conclusions and discussion

The WRF-Chem/WRFDA system is extended for the RACM-MADE-VBS-AQCHEM scheme (chem_opt = 109) to assimilate
surface $PM_{2.5}$ and $PM_{10}$ observations. The reliability and the effects of data assimilation using aqueous chemistry are demon-
strated through regional aerosol cycling where both PM measurements are assimilated along with conventional meteorological
data over the East Asian region every 6 h for one month from February 21, 2019.

By introducing aerosols in the aqueous (or cloud water) phase in WRFDA, the regional cycling system could represent wet
scavenging in resolvable-scale cloud microphysics and convective transport. The use of aqueous chemistry requires a double-
moment microphysics, for which the Morrison two-moment scheme is employed along with Grell-3 cumulus and the RRTMG
short- and long-wave schemes.

Both aqueous chemistry (AQ) and data assimilation (DA) systematically changed the atmospheric composition and its ver-
tical structure. Aerosol sizes get smaller when accounted for in clouds, and sulfate aerosols experience large increases in the
aqueous phase, in association with low-level clouds over the Korean peninsula. Although only surface PM concentrations are
assimilated, the impact goes up to the upper troposphere based on the background error statistics of each aerosol species. For
the period of 1-23 March 2019, DA with AQ increases PM concentrations with larger increments towards surface. Innovations
((o-b)'s) for categorized events show that DA_AQ slightly improves the fit to observations in all events except for the modest
underestimation in surface $PM_{10}$ in unhealthy (80 - 150 $\mu g\ m^{-3}$) and very unhealthy conditions (> 150 $\mu g\ m^{-3}$).

In a heavy pollution event in cloudy conditions, aqueous chemistry was particularly helpful in simulating wet deposition
of aerosols to accurately predict the evolution of surface PM concentrations in DA_AQ. As the activation, resuspension, and
wet scavenging processes can be all simulated only when aerosols in cloud water are defined through aqueous chemistry, DA
without aqueous chemistry treated all the aerosols as interstitial (e.g., suspended in the air) even when precipitation occurred,
leading to a significant overestimation of surface PM concentrations. At that time, large LWP was also produced over a wide



range of the domain, demonstrating that the formation and development of clouds were also affected by aerosols in the aqueous phase.

The use of aqueous chemistry in the aerosol cycling system is beneficial from several perspectives. First, major air pollution events over East Asia are often observed in association with significant cloud cover, where aerosols are considerably affected by clouds and have great impact on clouds as well. Second, cloudy conditions are generally hard to observe, especially in the remote-sensing data, to make data assimilation more challenging. In that case, model errors become more critical as the cost function will be largely dependent on the background error covariance. In the strong constraint variational data assimilation system where model errors are not taken into account, the solution (e.g., analysis) might be suboptimal when model errors become large. In other words, reducing model errors through sophisticated physics mechanisms can make data assimilation more effective as it violates the no-error assumption less significantly. Lastly, the effects of good-quality analysis can be accumulated to make systematic improvements in the air quality prediction by continuously cycling over a period of time.

Aqueous chemistry currently implemented in the WRF-Chem model is designed for warm-rain processes by treating aerosols only in the cloud-water phase. And aqueous chemistry implemented for the RACM-MADE-VBS scheme in WRF-Chem does not account for all the complex aqueous-phase reactions, either. As such, the aqueous chemistry used in the chemical option might be overly simple to represent all the physical processes for indirect aerosol effects, especially for mixed-phase convective clouds with nonprecipitating supercooled liquid water near cloud tops (Rosenfeld et al., 2008). In the case examined here, the new chemistry option was clearly helpful to simulate the reduction of PM concentrations due to wet removal of aerosol particles, associated with a fast-moving synoptic weather system. But in the cloudy conditions that do not result in precipitation, nor move fast, enhanced aerosol concentrations in the atmosphere can act to reduce the mean size of cloud droplets and suppress coalescence and warm-rain processes, while enhancing the growth of large hail and cold-rain processes. Those cases cannot be simulated in the model with such simple aqueous chemistry, which can mislead the analysis. As the strong-constraint 3D-Var system used in this study does not include any model error term, model errors are not investigated nor discussed in detail, but there is room for further improvements for the RACM-MADE-VBS-AQCHEM option in the model to account for all the aerosol effects in clouds and precipitation. Also, the particular scheme hardly includes any diagnostics variables for each aerosol species to make it hard to examine the physics mechanism behind the final product (e.g., surface PM concentrations). It would be good to add some diagnostics in the scheme to better understand the production by $SO_2$ aqueous-phase oxidation or wet deposition of different aerosol species, for instance.

On the other hand, in the WRFDA system developed for this study, aerosol number concentrations are not included as part of analysis (or control) variables so that they are not changed through the assimilation. To fully describe aerosol impacts on clouds or to handle complex cases with mix-phased clouds or cold-rain processes, we might need to consider developing the assimilation system to reflect the changes in aerosol number concentrations per aerosol (size) mode. Recently, cloud properties and/or atmospheric constituents are increasingly measured or derived from multiple platforms on the ground and space-born satellites. Needless to say, those data can be extremely valuable to not only evaluate the modeling system but also to better initialize the model through data assimilation.



*Code and data availability.* The WRF V4.3.1 codes are freely available from https://www2.mmm.ucar.edu/wrf/users/download/get_source.html (NCAR, 2022). The WRFDA source codes developed for this study can be downloaded from https://doi.org/10.5281/zenodo.4594671 (Ha, 2022). Real-time air quality observations are available at http://www.airkorea.or.kr/ (Air Korea, 2022) for Korean sites and at http://www.cnemc.cn (China National Environmental Monitoring Centre, 2022) for Chinese stations. The National Centers for Environmental Prediction (NCEP) PREPBUFR format data are archived and available at Research Data Archive at the National Center for Atmospheric Research, Compu-

tational and Information Systems Laboratory. https://doi.org/10.5065/Z83F-N512. The WRF-Chem preprocessor tools such as mozbc and megan_bio_emiss are available at https://www.acom.ucar.edu/wrf-chem/download.shtml (ACOM/NCAR, 2022).

*Competing interests.* The author has declared that there are no competing interests.

*Acknowledgements.* This research is based upon work supported by the National Center for Atmospheric Research which is a major facility sponsored by the National Science Foundation under Cooperative Agreement No. 1852977. The code development was also supported in part by a grant from the National Institute of Environment Research (NIER), funded by the Ministry of Environment (MOE) of the Republic of Korea (NIER-2021-01-02-086). The author acknowledges the use of the WRF-Chem preprocessor tools (mozbc and megan_bio_emiss) provided by the Atmospheric Chemistry Observations and Modeling Lab (ACOM) of NCAR. Big thanks go to Dr. Duseong Jo at ACOM/NCAR

for his internal review and fruitful discussion. All the simulations, data processing and analysis were conducted on NCAR's supercomputer.

*Author contributions.* Ha developed the codes, designed the experiments, analyzed the results, and prepare the manuscript.



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



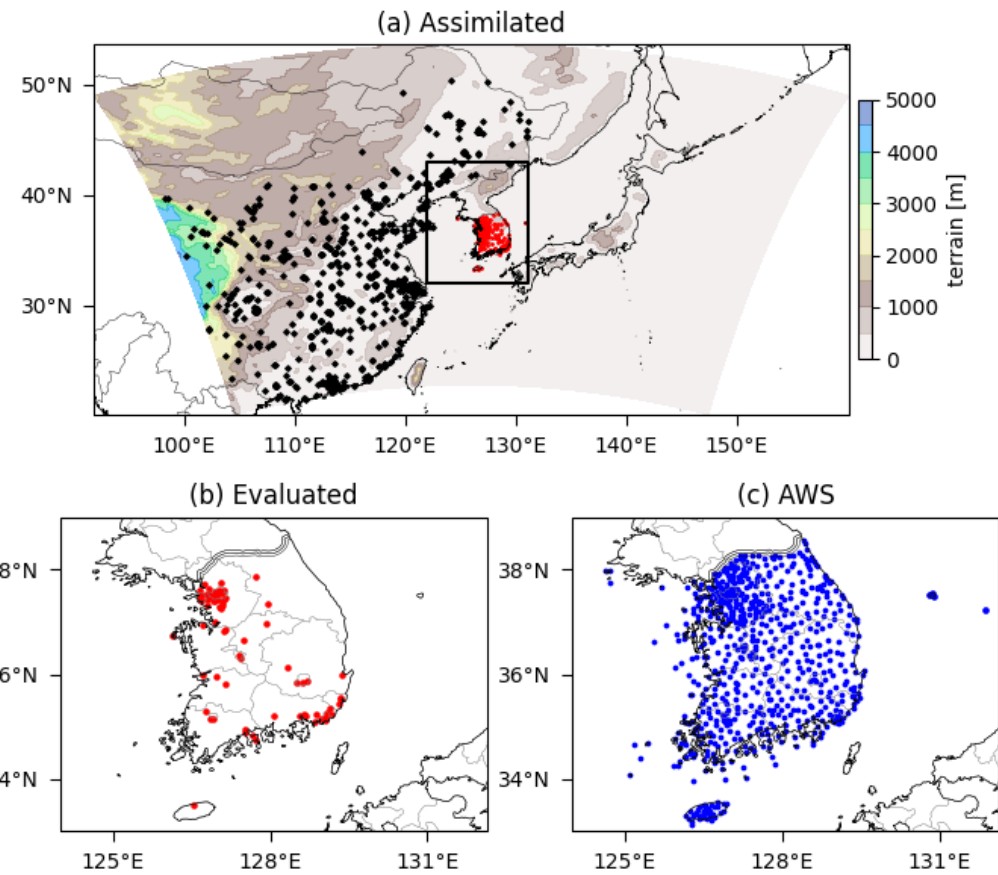

**Figure 1.** Surface observation network used in this study: a) Assimilated observation sites are marked as dots in red (black) for Korean (Chinese) sites, with terrain height colored in domain 1 (at 27 km resolution) and a black box over the Korean peninsula indicating domain 2 (at 9 km grid resolution) b) the sites for evaluation and c) the Korean automated weather stations (AWS) measuring meteorological variables at the surface.





**Figure 2.** Vertical profile of each aerosol species in background error standard deviation estimated with and without aqueous chemistry (AQ and NO_AQ, respectively) over domain 2 (D2). Accumulation mode aerosols in AQ (NO_AQ) are depicted in red lines with dots (black solid lines) while Aitken mode aerosols in AQ (NO_AQ) in dashed orange (dotted gray) lines.





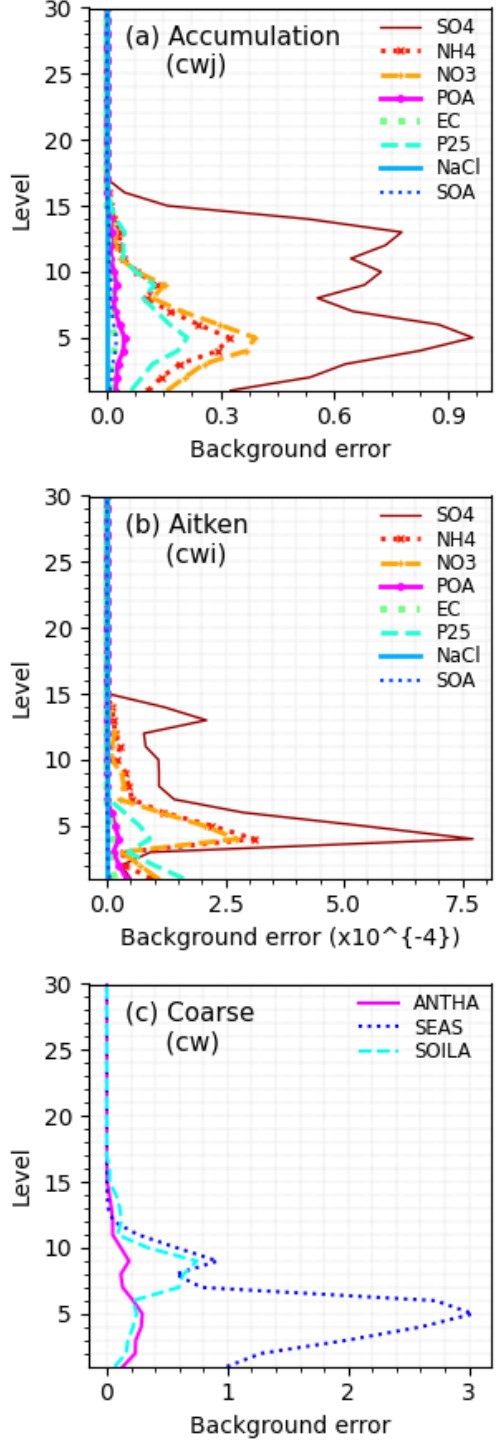

**Figure 3.** Vertical profile of background error standard deviation with aqueous chemistry (AQ) for aerosol species in the aqueous or cloud water ("cw") phase (D2).

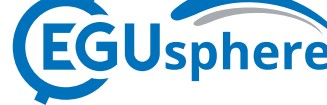

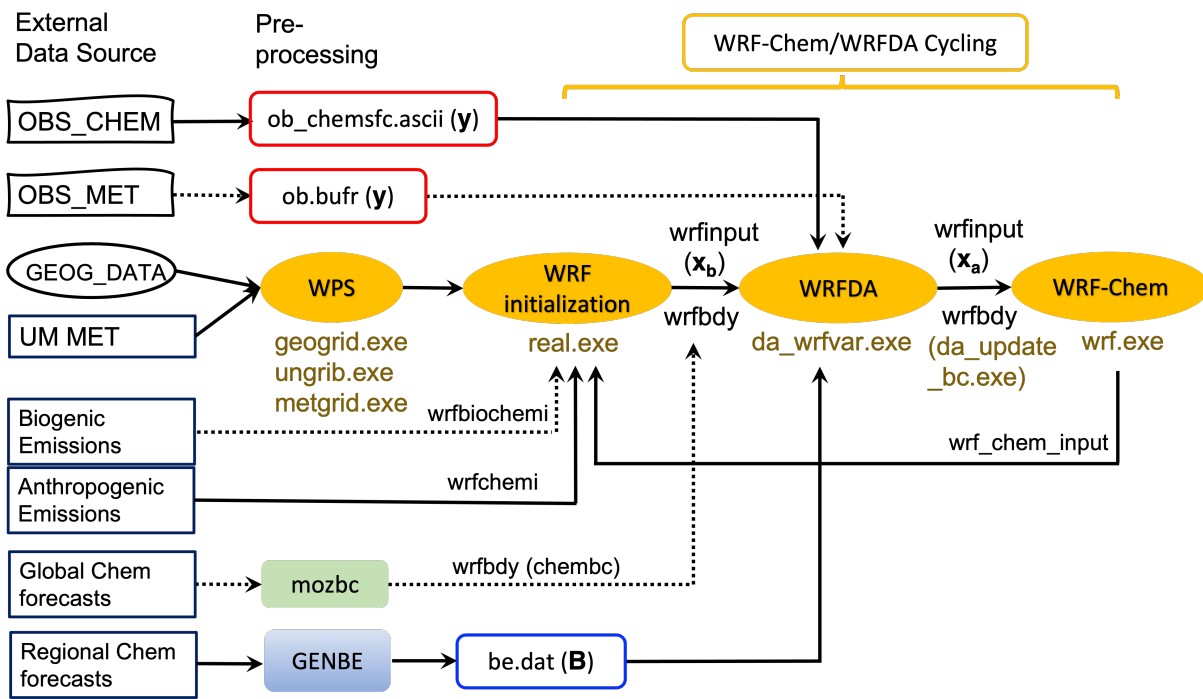

**Figure 4.** Flowchart of the WRF-Chem/WRFDA cycling system for chemical data assimilation. Gridded input data are indicated by rectangular boxes on the left and all the software packages are filled in colors. Dotted lines imply optional input data while solid lines the mandatory inputs for WRF-Chem/WRFDA cycling, accompanied by typical input file names (with no specification of domain ID and time) used in the WRF system.





**Figure 5.** Vertical profile of each aerosol species ($[\mu g\ kg^{-1}]$) in the analysis and background (in D2) with and without aqueous chemistry (DA_AQ and DA, respectively), averaged over 71 verification sites in Korea from 6-hr cycling for 1-23 March 2019.



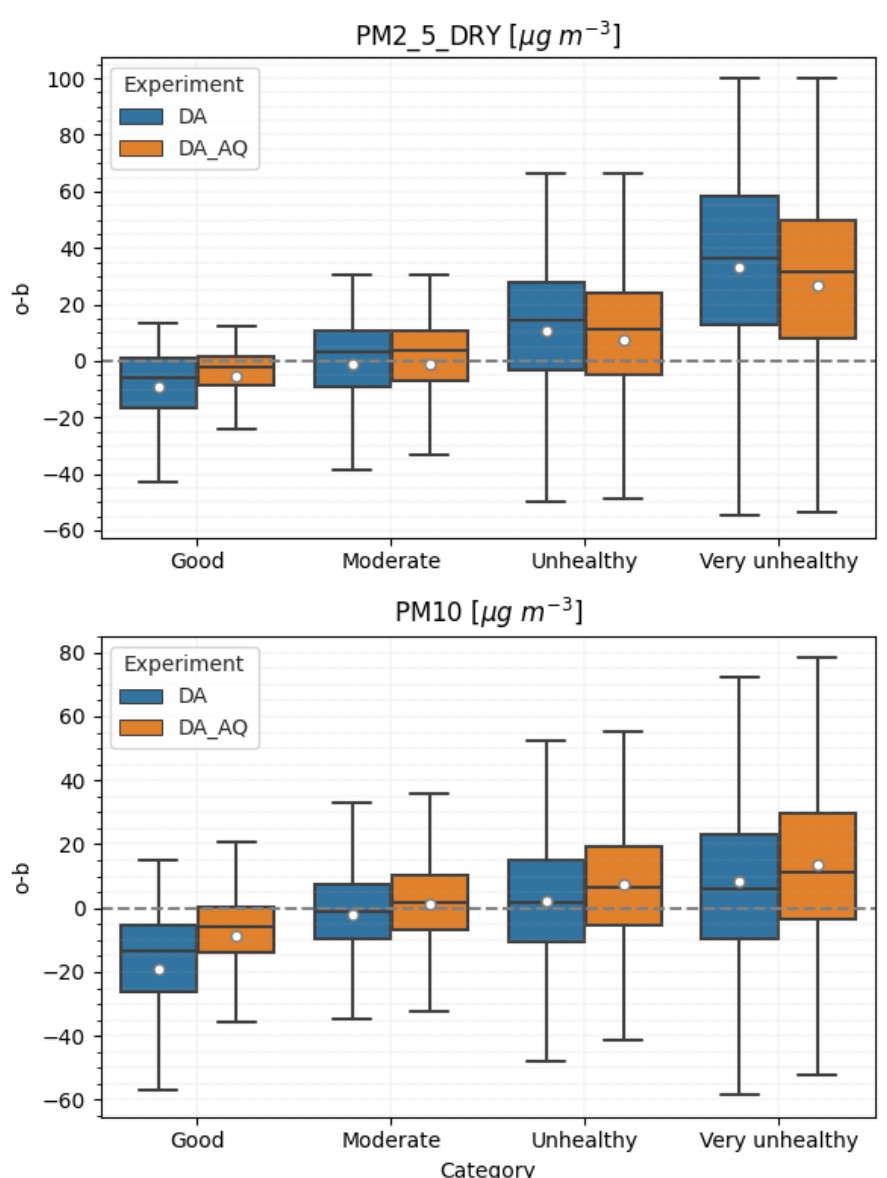

**Figure 6.** Box plot of innovations((o-b)'s) in each category for (top) PM$_{2.5}$ and (bottom) PM$_{10}$ concentrations on the ground, comparing DA and DA_AQ in domain 2 from 6-h cycling for 1-23 March 2019. White dot in each box is the mean deviation of 6-h forecasts ("b") from the observations ("o"), while the black horizontal line in the middle of each boxplot stands for the median of (o-b)'s. Observations ("o") used here are marked as red dots in Fig. 1a.

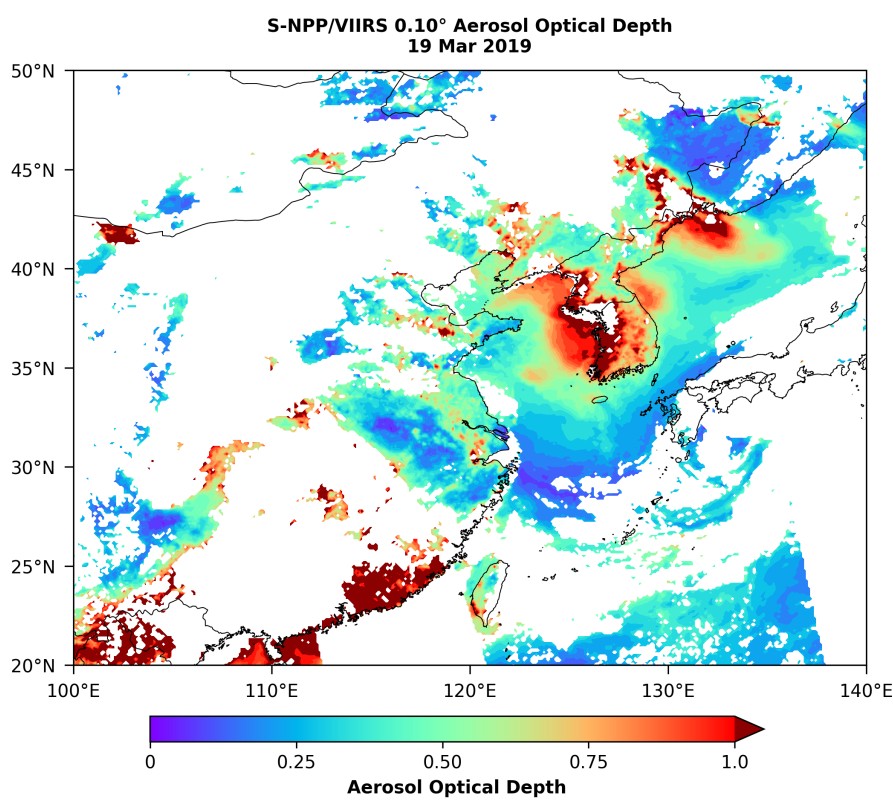

**Figure 7.** Aerosol optical depth retrieved from VIIRS onboard the Suomi NPP as a daily mean gridded data (level 3) on Mar 19, 2019.



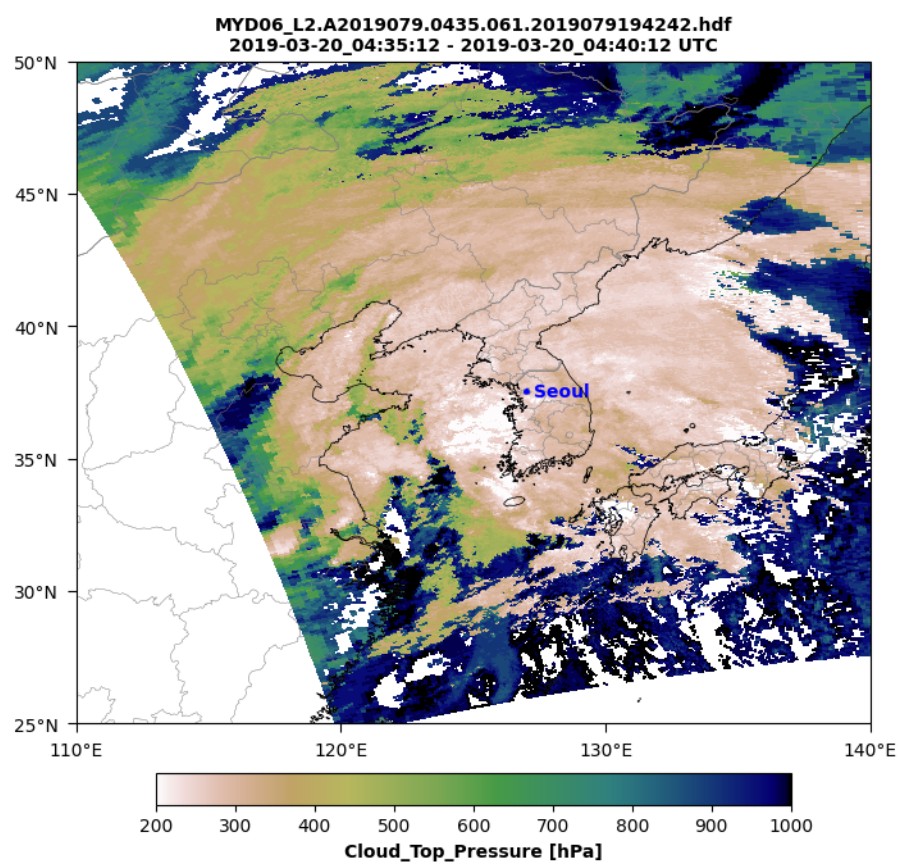

**Figure 8.** Level 2 cloud top pressure retrieved from the MODIS sensors onboard the Aqua satellite, merged between 04:35:12 and 04:40:12 UTC on 20 March 2019.

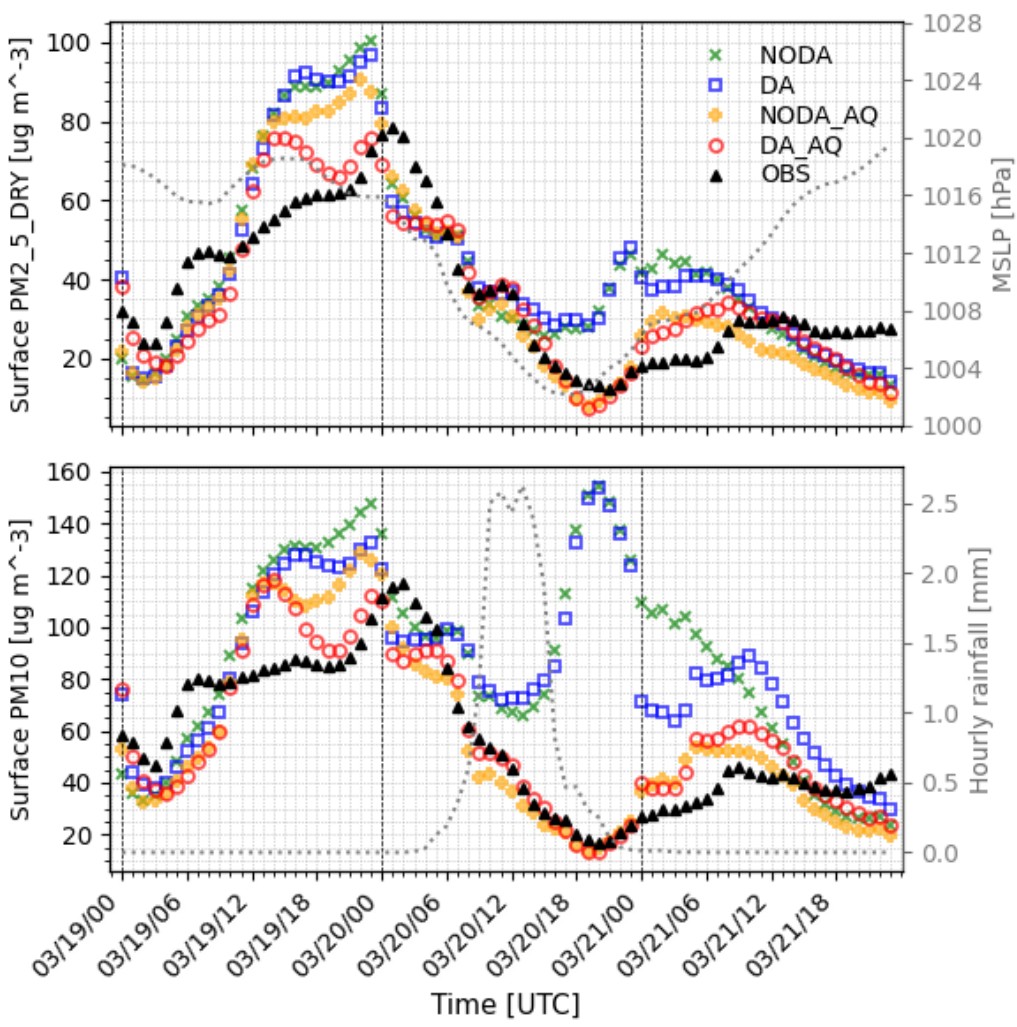

**Figure 9.** Time series of (top) surface $PM_{2.5}$ and (bottom) $PM_{10}$ concentrations for three days (19-21 March 2019), averaged over 71 Korean verification sites (marked in Fig. 1b). In-situ observations (OBS; black triangle) are compared with cycling experiments for their 0-23 h hourly forecasts from the 00Z analysis every day. Gray dotted lines with the right y-axis are (top) mean sea level pressure (hPa) and (bottom) hourly rainfall (mm) observations averaged over 699 AWS sites over South Korea (marked in Fig. 1c).





**Figure 10.** Time series of 0-24 h forecasts of PM$_{10}$ concentrations (colored) simulated at Seoul, South Korea (in model levels up to 20) in each experiment. Cloud and Rain mass mixing ratios (QCLOUD and QRAIN ([g kg-1])) are contoured in white and pink, respectively. Liquid Water Path (LWP) is also plotted in dashed lines with the right y-axis.





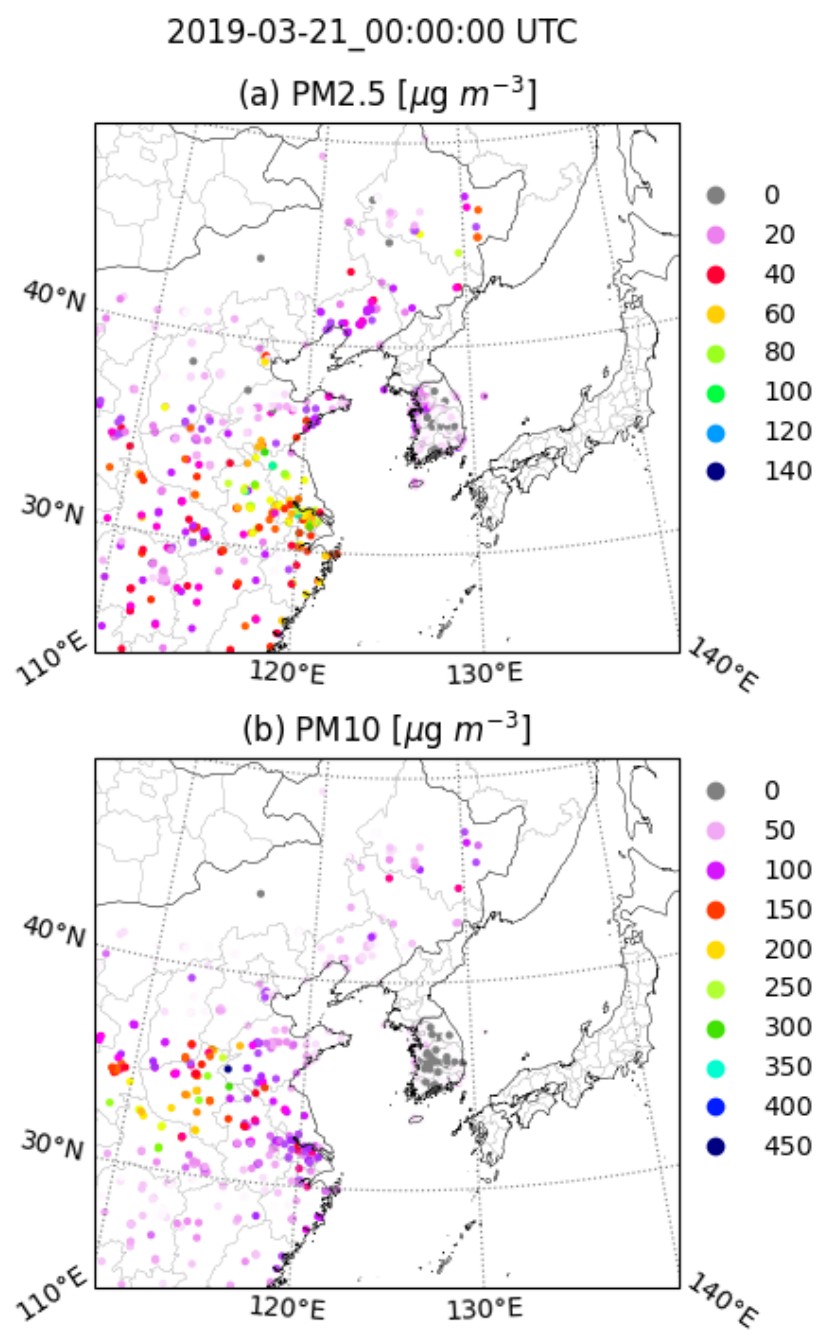

**Figure 11.** Surface observations for (a) PM$_{2.5}$ and (b) PM$_{10}$ concentrations valid at 2019-03-21_00:00:00 UTC.





**Figure 12.** Horizontal distribution of PM$_{2.5}$ concentrations simulated at the lowest model level in 24 h forecast valid at 2019-03-21_00:00:00 UTC in each experiment.





**Figure 13.** Same as Fig. 12, but for PM$_{10}$.




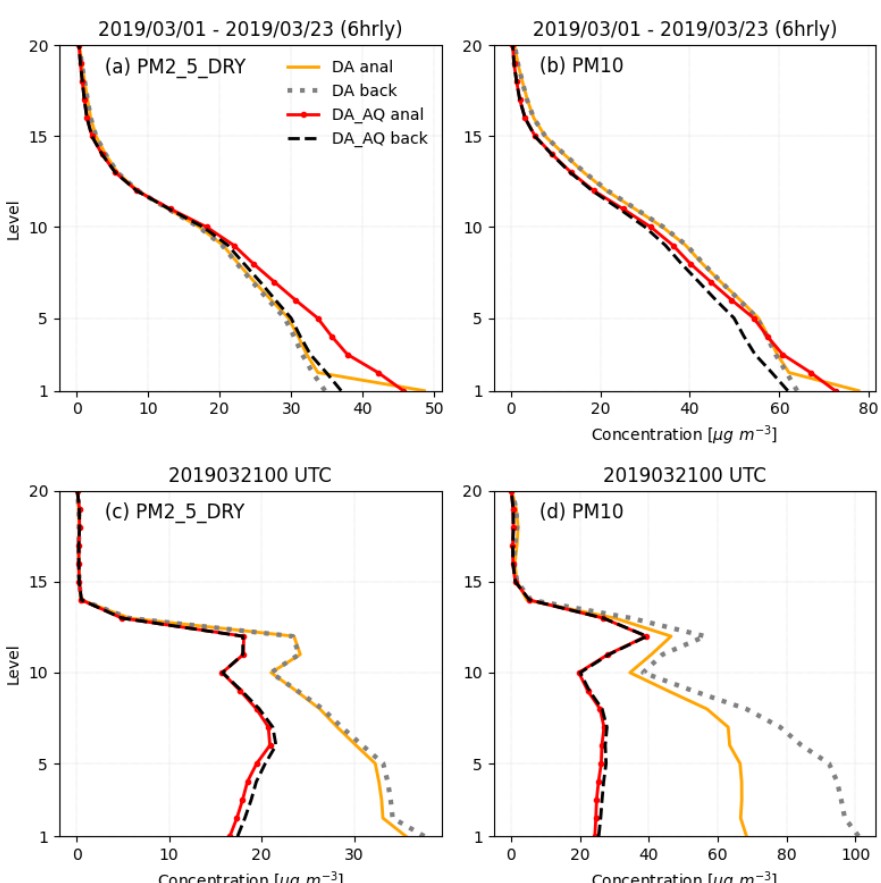

**Figure 14.** Vertical profile of PM concentrations ([$\mu$g kg-1]) in the analysis and background with and without aqueous chemistry (DA_AQ and DA, respectively) over domain 2, averaged over 71 verification sites in South Korea. The upper panel displays the time mean from 6-hr cycling for 1-23 March 2019 and the bottom panel for the cycle at 00 UTC 21 March 2019.




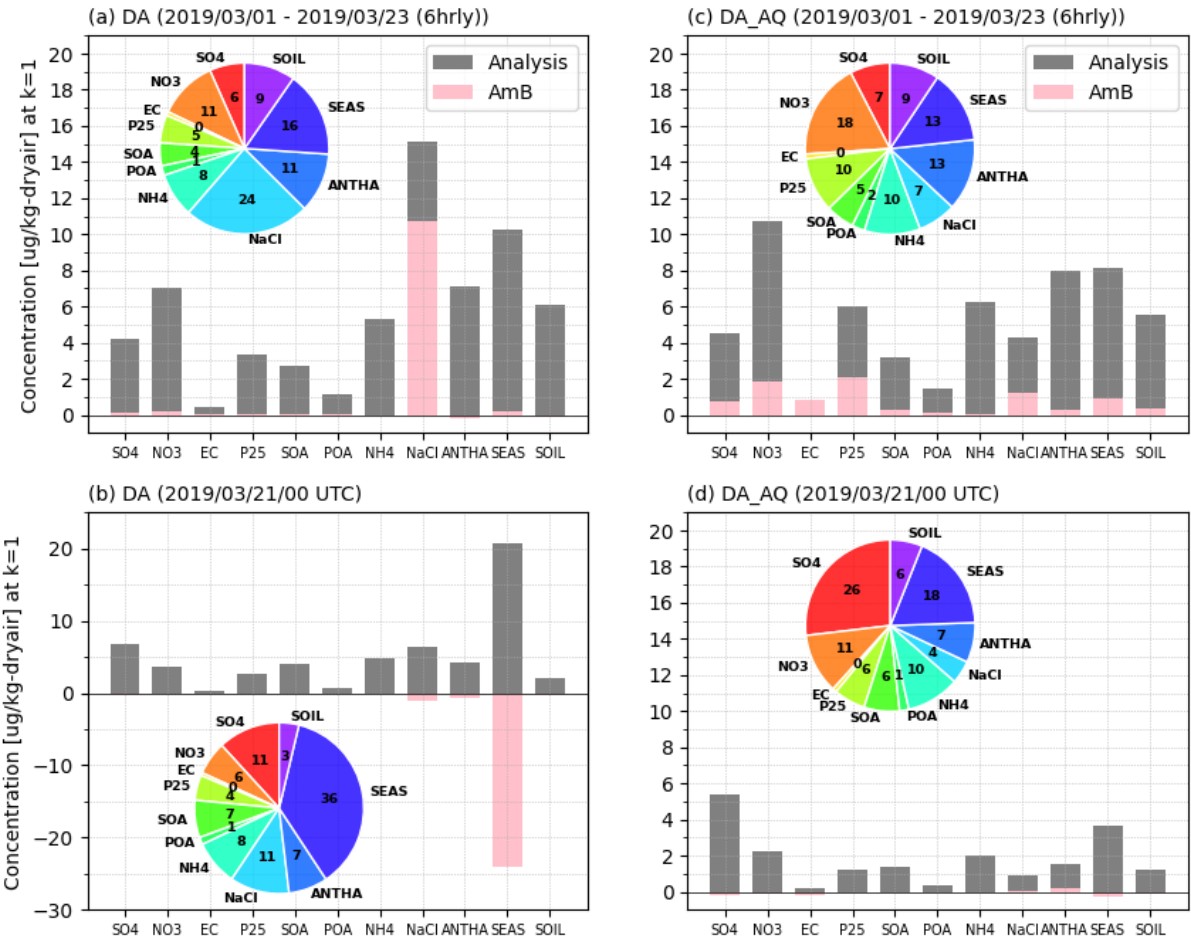

**Figure 15.** Bar graph compares concentration of each aerosol species averaged over 71 verification sites over South Korea at the lowest model level. Left panels present analysis (gray) and analysis increment (AmB; pink) in DA, while right panels show DA_AQ experiments. Top panels show the cycle mean for Mar 1-23, 2019 and bottom panels on the analysis at 00 UTC 21 March 2019 (bottom). Pie chart illustrates the atmospheric composition in the analysis.



**Table 1.** Physics and chemical options

| Scheme | Option |
| --- | --- |
| Microphysics | Double-moment Morrison |
| Longwave radiation | RRTMG |
| Shortwave radiation | RRTMG |
| Surface layer | Monin–Obukhov (Janjic) |
| Land surface | Noah |
| Boundary layer | YSU |
| Cumulus parametrization | Grell-3D |
| Chemistry driver | RACM |
| Aerosol driver | MADE-VBS |
| Biogenic emissions | Gunther |
| Gas chemistry | On |
| Aerosol chemistry | On |
| Cloud chemistry | On |
| Aqueous chemistry | On |
| Aerosols – cloud – radiation interactions | On |
| Wet scavenging | On |
| Vertical mixing | On |

**Table 2.** Experiments. DA stands for data assimilation and AQ aqueous chemistry.

| Experiment | chem_opt | Assimilation |
| --- | --- | --- |
| NODA | 108 | None |
| DA | 108 | CHEM+MET |
| NODA_AQ | 109 | None |
| DA_AQ | 109 | CHEM+MET |

**Table 3.** Air quality index values, as defined in the NIER in Korea.

| Concentration | Good | Moderate | Unhealthy | Very Unhealthy |
| --- | --- | --- | --- | --- |
| $PM_{2.5}$ [$\mu g/m^3$] | 15 | 35 | 75 | > 75 |
| $PM_{10}$ [$\mu g/m^3$] | 30 | 80 | 150 | > 150 |