# Peer review of "Aerosol data assimilation with aqueous chemistry in WRF-Chem/WRFDA V4.3.1"

_EGUsphere, 2022_

## Referee Comment (RC1)

**General Comments:**

This paper introduces a new chemistry option within the WRFDA system to activate the aqueous chemistry for the assimilation of surface concentrations of particulate matter. The proposed algorithm is tested with cycling data assimilation experiments, which presents improvements in the simulation of aerosol wet removal processes and atmospheric composition. However, more analyses should be provided to support the drawn conclusions in this paper. Therefore, a major revision is recommended based on the following comments.

**Specific Comments:**

1. The primary issue remain in the paper is the missing of the forecast verification on the entire experimental period. A month-long experiment with cycling data assimilation conducted in this study provide abundant samples for the verification on the forecast. However, the forecast verification is only concentrated over three days (i.e. Fig. 9), and over a specific time (i.e. Figs. 11-13). The results using the one-month experiment are just for the analysis field (i.e. Figs. 5 and 14). To draw a solid conclusion for the impact of the introduced chemistry option on the forecast, verifications on the month-long experiment should be provided.

2. A total of 31 vertical levels is used in the experiments in this study. It is noted that the setting was to follow the configuration at the NIER, while such an explanation may not be strong enough for not using a higher vertical resolution to perform experiments in this study. Also, as descripted around L405, "the poor quality of PM forecasts over nighttime is in part because of the coarse vertical resolution employed in this study". Therefore, it is recommended to perform an experiment with higher vertical resolution, at least for the case verification over several days.

3. The WRF-Chem/WRFDA system provides a useful approach to assimilate aerosol/meteorological observations and to get the forecasts. A detailed description about the WRF-Chem for aerosol effects is given in section 2. By contrast, the description about aerosol assimilation within WRFDA is only mentioned with several sentences in the

beginning of section 3. It is better to present a more detailed background for the aerosol assimilation within WRFDA.

4. Some minor comments are summarized as below.

1) Please provide further analysis to support the statement around L363, "implying that the spatial variability of surface PM concentrations goes up with haze events that might not be fully captured by the 9-km simulations".

2) Please add results based on observations in Fig. 10, such as the observations of  $PM_{10}$  and precipitation.

3) To support the changes in the vertical profiles as in Figs. 5 and 14, it is better to provide some vertical profiles based on available observations.